# Trawl Fisheries in the Gulf of Thailand: Vulnerability Assessment and Trend Analysis of the Fish Landings

Pavarot Noranarttragoon [1], Sontaya Koolkalaya [2], Weerapol Thitipongtrakul [1], Piyathap Avakul [3], Ratanavaree Phoonsawat [4] and Tuantong Jutagate [5,*]

[1] Marine Fisheries Research and Development Division, Department of Fisheries, Bangkok 10900, Thailand; pavarotn@gmail.com (P.N.)
[2] Faculty of Agricultural Technology, Rambhai Barni Rajabhat University, Chanthaburi 22000, Thailand
[3] Academic and Curriculum Division, Mahidol University, Nakhonsawan Campus, Nakhonsawan 60130, Thailand
[4] Office of Fisheries Expert, Department of Fisheries, Bangkok 10900, Thailand
[5] Faculty of Agriculture, Ubon Ratchathani University, Ubon Ratchathani 34190, Thailand
[*] Correspondence: tuantong.j@ubu.ac.th; Tel.: +66-45-353500

**Abstract:** Vulnerability of each stock in the catches from trawl fisheries in the Gulf of Thailand was assessed by productivity susceptibility analysis. Separate assessments were made based on type of trawler (otter-board, pair and beam) and vessel size (SS, S, M, L and XL, according to gross tonnage). Catches were distinctly different among trawler types and were dominated by demersal fishes, pelagic fishes and shrimps in the otter-board, pair and beam trawlers, respectively. The stocks of over 200 taxa were included in the study; high-vulnerability taxa were found for all trawl types and sizes, except the beam trawler size S. Only seven (7) taxa were classified as high vulnerability, including four (4) teleost taxa, namely *Saurida elongata*, *Plotosus* spp., *Gymnothorax* spp. and *Sphyraena* spp., and three (3) elasmobranch taxa, namely *Carcharhinus* spp., *Brevitrygon heterura* and *Neotrygon kuhlii*. Meanwhile, as many as 26 high-medium vulnerability taxa were found in catches by otter-board trawler size L, which included not only fishes but also cephalopods. Trends and variation in fish landings of 20 high and high-medium vulnerability taxa were analyzed. Eighteen (18) taxa showed monotonic trends, both continuous and discontinuous, in their time series of annual landings, but not *Muraenesox* spp. nor *Uroteuthis* spp. The long-term variations in fish landings ranged mostly between 40% and 50%. The short-term variations showed positive absolute and relative skewness, i.e., mostly between 10% and 20%. Results of this study highlight the taxa that require a precautionary approach for fishery management and warrant comprehensive fish stock assessment. Such data will allow more accurate determination of their status compared to the reference points and facilitate better management of trawl fisheries in the Gulf of Thailand.

**Keywords:** catch composition; productivity and susceptibility analysis; monotonic trend; long-term variation; short-term variation

**Key Contribution:** This study identifies the potential risk of trawl fisheries in the Gulf of Thailand regarding their catches. Trends in fish landing of some high and high-medium vulnerability species are also presented. Results can be further used to identify the species that require specific stock assessment and mitigation to ensure that they are sustainably managed.

## 1. Introduction

Fishery production in Thailand was about 2.62 million tons in 2020, of which 56% (1.47 million tons) was from marine fisheries [1]. The marine fisheries in Thailand are multi-species, and the impacts from fishing are considerable not only for the targeted species but also non-targeted species. The rate of discarding from the catch is considerably low, however (estimated at only 1%), because the low-value species are typically utilized to

produce fish sauce and fish meal [2–4]. Impacts of fisheries on fish stocks, as well as the integrity of aquatic ecosystems and habitats, are among the top concerns of the Department of Fisheries (DoF), Thailand. To mitigate these impacts, the marine fisheries in Thailand have been moving toward a more sustainable approach, by embracing ecosystem-based fisheries management. Strict regulations have been implemented to reduce overcapacity of fleets and to control catches and effort since 2015, when the new Royal Ordinance on Fisheries was declared. Moreover, DoF follows Sustainable Development Goal 14 of the United Nations (UN-SDG 14: Life Below Water) by striving to effectively regulate harvest and end illegal fishing and overfishing, as well as implement science-based management plans [5,6].

A large proportion (about three-fourths) of the marine catches in Thailand is from the Gulf of Thailand (GoT). The GoT is among the most productive bodies of water in the world, due to its supply of nutrients from many large rivers (e.g., the Chao Phraya and the Mekong) and large, biologically diverse marine ecosystems. Furthermore, ca. 63% of the Gulf's surface area (320,000 km$^2$) is in Thai territory [7]. The GoT is relatively shallow, having a mean depth of 45 m and maximum depth of only 80 m; the circulation is generally weak and variable [5,7]. A wide variety of fishing gears are operated in the GoT, including traditional gears (e.g., mullet gill nets, shrimp gill nets and crab traps) for small-scale fisheries as well as highly efficient gears (e.g., falling nets, purse seines and trawls) for commercial fisheries [5,6]. The DoF divides the trawl fisheries into three gear types, namely otter-board trawl, pair trawl and beam trawl, which are further divided into five categories according to the size of the vessel: SS (<10 gross tonnage, GT), S (10–29.9 GT), M (30–59.9 GT), L (60–149.9 GT) and XL (>150 GT). Among the trawls, the otter-board is currently the most popular; in 2021, the number of registered otter-board vessels was 1812. In the same year, the numbers of registered pair trawls and beam trawls were 1124 and 434, respectively [8]. Supongpan and Boonchuwong [9] reported that the fishing grounds of small otter-board trawlers are within 20 m depth, and near shore, medium size trawlers are within 30 m depth in the coastal zone, while pair trawlers are in deeper waters, up to 50 m. Suuronen et al. [10] also mentioned that the maximum depth of trawling in Southeast Asian (SEA) countries is about 70 m or less, mostly because of poor technical capacity of vessels. Tossapornpitakkul et al. [11] reported that otter-board trawlers operated mostly during nighttime and that the catch rates during the inter-monsoon period were higher than during the monsoon seasons. In contrast, the pair and beam trawlers are normally operated year-round in daytime [9].

Similar to other SEA countries, a large proportion of the marine harvest in Thailand is from the trawl fisheries [10,12]. Within the Thai portion of the GoT, catches from trawl fisheries are normally about half of the total annual catch, for example: 0.47 out of 1.04 million tons in 2019 and 0.42 out of 0.96 million tons in 2020 [1,13]. The recently estimated catches from the GoT in 2020 showed that, among the three trawl types, the majority of the catch came from pair trawls (272,265 tons), followed by otter-board and beam trawls, which accounted for 135,532 and 12,657 tons, respectively [1]. Management of the trawl fisheries in Thai waters is implemented by a suite of measures: (a) gear regulation, whereby mesh size of the cod-end must be larger than 4 cm stretched mesh, (b) fishing closures based on season and area, where areas less than 3 nautical miles are totally closed to trawl fisheries, and (c) license and effort control, for which the allowable catch and effort for the trawl fisheries were at 756,076 tons and 16.25 million hours, respectively, between 2019 and 2021 [5]. Moreover, monitoring of potential impacts of trawling on seabed habitats and biota is conducted by the Department of Marine and Coastal Resources [6].

With a large number of species in the catch, understanding stock status of individual species and their vulnerability to harvest is important for management and setting appropriate reference points, both for economically important species and those of special concern or at risk [14]. Given that numerous aquatic animal taxa are harvested by tropical trawl fisheries, the data necessary to cover all the exploited stocks for intensive fish stock assessment are always insufficient [12,14–16]. Alternatives to the risk assessment of

individual exploited stocks have therefore been introduced and recommended, such as the semi-quantitative "Productivity-Susceptibility Analysis (PSA)" [12,15,17,18]. This method per se represents the second of three levels in a hierarchical ecological risk assessment framework for quantifying the effect of fishing, in which a comprehensive set of attributes that are proxies for productivity and susceptibility have been identified for evaluating vulnerability of a species in a particular fishery [16,17]. Due to its usefulness and effectiveness in rapidly screening the degree of impact of a fishery on individual taxa, the PSA has been recommended for many data-poor and mixed-species fisheries in Asia [12,15,18]. In Thailand, this method has been applied to understanding the impact of fisheries to seahorses [19] and for examining the catch from small-scale fisheries for blue swimming crab [20].

The trawl fisheries are an important supplier of seafood in Thailand, where the catches are largely used to support domestic seafood consumption and processing, as well as the fishmeal industry. The Thai Sustainable Fisheries Roundtable (TSFR), by cooperating, has recently announced their partnership to work towards responsible trawl fisheries. The TSFR has applied the international standards for trawl fisheries in the Gulf of Thailand and in November 2020 became the first Fishery Improvement Project (FiP) for mixed-species fisheries [21]; the current phase of this project involves scientific studies on the impacts of trawling on fish stocks and habitats. A recent multispecies fisheries assessment [22] found that although the management objectives and reference points are set at a fair level, the management by groups, i.e., anchovies, pelagic fishes and demersal species (5, 13), raises concern about risk to individual species in the catch. This paper, therefore, aims to (i) apply PSA to evaluate the vulnerability of individual species (or group of species) from catches by each trawl type and size and (ii) examine trends and variation in fish landings of the high or high-medium vulnerability taxa from trawl fisheries in the GoT. The results provide a baseline for risk levels and trends of individual taxa and can be further used to identify the species that may require specific stock assessment and mitigation to ensure that they are sustainably managed.

## 2. Materials and Methods

### 2.1. Catch Data

The study used data from surveys of fish landings around the GoT, conducted by five regional centers of the Marine Fisheries Research and Development Division (MFRDD) between 2016 and 2020, which were pooled by the Fisheries Resource Assessment Group of MFRDD. Each landing site was visited once a month, and data from all the "port-in" trawlers of that day were collected. Because of the unpredictable "port-in" time of the trawlers, it was difficult to predict beforehand which types of trawlers would be included in each sampling event. This was particularly true for the trawlers of size M and above, for which the operation period varies from a few days to 15 days, depending on weather and size of catches. Number of the sampling trips for the otter-board trawlers were 39, 55, 52 and 58 for the vessel sizes S, SS, M and L, respectively. Meanwhile, there were 38, 111 and 12 sampling trips for the pair trawlers of size M, L and XL, respectively. Lastly, there were 34, 36 and 24 sampling trips for the beam trawlers of size S, M and L, respectively. Collected information included fishing effort, fishing ground and total catch. The catch of each trawler was sorted into taxonomic groups, by the well-trained fishery scientists from each center, based on FishBase (www.fishbase.org (accessed on 18 May 2022); [23]) for fishes and SeaLifeBase (www.sealifebase.org (accessed on 18 May 2022); [24]) for other aquatic animals, and then weighed (to 0.1 kg). The conservation status of each taxon was listed, according to the IUCN Red List of Threatened Species (www.iucnredlist.org (accessed on 18 May 2022); [25]). Only the portion of the catch that was identified to genus or species level was used in the PSA. Another set of data, i.e., fish landing data, used in trend and variation analyses, was acquired from fisheries statistics of Thailand, which are annually reported by the Fisheries Statistics Group, Fisheries Development Policy and Planning

Division, DoF; these data are available from 1985 to 2020 (https://www4.fisheries.go.th/local/index.php/main/site/strategy-stat (accessed on 18 May 2022)).

### 2.2. Productivity and Susceptibility Analysis (PSA)

Productivity Susceptibility Analysis (PSA; [17]), which is a practical semi-quantitative vulnerability assessment tool [16,26,27], was used for assessing the risk of individual stocks from the trawl fisheries in the GoT, according to the trawl type and vessel size. The PSA incorporates the attributes of two characters, i.e., productivity and susceptibility. The productivity attributes are employed to determine the recovery rate of a stock from fishing, while the susceptibility attributes are used for determining the extent of the impact of the fishery on individual stocks [16,17]. Seven productivity attributes and four susceptibility attributes were used in this study (Table 1, [12]). For each species, the information for each attribute was from sampling (i.e., contribution in catch) and from desk study of relevant scientific publications, in particular from MFRDD (www.fisheries.go.th/library, (accessed on 18 May 2022)) as well as from FishBase (www.fishbase.org (accessed on 18 May 2022); [23]) and SeaLifeBase (www.sealifebase.org (accessed on 18 May 2022); [24]) using common and scientific names as search keywords. The obtained information was converted to a rank score (Table 1), where 1 is high productivity or low susceptibility; 2 is medium productivity or medium susceptibility; and 3 is low productivity or high susceptibility [16]. It is worth noting that some taxa were difficult to identify to species level in the field; these were reported as groups of species, i.e., at genus level. A focus group (an important early step of PSA) among the researchers, fishery scientists, trawl fishers and representatives of the fish meal group and TSFR was assembled to discuss and agree upon the rank scores of the stocks, based on their experience and ecological knowledge. The total vulnerability (V) or risk score was then calculated as the Euclidean distance from the origin, which allows a single risk ranking, by

$$V = \sqrt{P^2 + S^2}$$

where P is the overall productivity score (i.e., arithmetic mean of the productivity attributes), and S is the overall susceptibility score (i.e., geometric mean of the susceptibility attributes). The V-score ranges between 1.41 and 4.24; values lower than 2.64 and above 3.18 are considered low and high vulnerability, respectively, while values in between indicate medium vulnerability [16,17]. In addition, in this study, the stocks receiving a V-score between 3.00 and 3.17 are considered as having high-medium vulnerability. Data quality of the inputs was rated as 1: data collected from the area of interest; 2: information from the literature, FishBase and SeaLifeBase of other region; 3: as in 2, but at the genus or family level; 4: expert opinion or 5: no data. The mean quality score of P and S was interpreted as high (<2), medium (≥2 and <3) or low (≥3) [20,25,28].

### 2.3. Statistical Analysis

The difference in V-score by group of aquatic animals according to each vessel size and trawl type was tested by the Kruskal–Wallis test, i.e., one-way ANOVA on ranks, and Dunn's post test was applied if a significant difference was found at $\alpha = 0.05$. Trends in catch of 20 high and high-medium vulnerability taxa were examined by Spearman's rank correlation to detect monotonic trends in time series of landings [29]. Discontinuity between trends was examined by the most significant turning point, which resulted in the maximum weight rank ($r_s^2$) as

$$r_s^2 = \left( n_1 r_1^2 + n_2 r_2^2 \right) / n$$

where $n_1$ and $n_2$ are number of years in the first and second sub-series, and $r_1$ and $r_2$ are Spearman's rank correlation for the first and second sub-series, respectively [29].

The variation in 20 selected taxa was expressed by coefficient of variation (CV0) in fish landings. Furthermore, to investigate the long-term variation, the coefficients of variation in the 1st (linear trend, CV1) and 2nd order polynomials (CV2) were applied by using

time (years) and fish landings as predictor and response, respectively [30,31]. Short-term variation (i.e., percent change in fish landings from one year to the next) was examined by indices of absolute variation ($U_a$) and relative variation ($U_r$) as

$$U_a = 100 \times \frac{\text{mean}\left|y_i - y_{i-1}\right|}{y} \ (\%)$$

$$U_r = 100 \times 2 \times \left(\left(1 - \frac{1}{10^r}\right) \Big/ \left(1 + \frac{1}{10^r}\right)\right) \ (\%)$$

where y = mean of fish landings (tons) from long-term data; $y_i$ and $y_{i-1}$ = fish landings in a given year and previous year, respectively; and r is the mean of absolute difference of log-transferred fish landings as calculated by

$$r = \sum_{i=2}^{n} \left|\log_{10}(y_i/y_{i-1})\right| \Big/ (n-1)$$

where n = duration of time series data (years) [30]. Significant difference between $U_a$ and $U_r$ was tested by the two-sample *t*-test. All statistical tests were conducted by using R [32].

**Table 1.** Attributes and scoring thresholds used to determine the vulnerability of taxa caught by trawl fisheries in the Gulf of Thailand.

| (A) Productivity Attributes and Rank Score | | | |
|---|---|---|---|
| **Productivity Attributes (P)** | **Low Productivity/ High Risk** | **Medium Productivity/ Medium risk** | **High Productivity/ Low Risk** |
| | **Rank Score = 3** | **Rank Score = 2** | **Rank Score = 1** |
| P1: Average age at maturity (years) | >4 | 2–4 | <2 |
| P2: Average maximum age (years) | >30 | 10–30 | <10 |
| P3: Fecundity (eggs/spawning) | <1000 | 1000–10,000 | >10,000 |
| P4: Average maximum size (cm) | >150 | 60–150 | <60 |
| P5: Average size at maturity (cm) | >150 | 30–150 | <30 |
| P6: Reproductive strategy | Live bearer, or significant parental investment | Demersal spawner | Broadcast spawner |
| P7: Mean trophic level | >3.25 | 2.5–3.25 | <2.5 |
| **(B) Susceptibility attributes and rank score** | | | |
| **Susceptibility attributes (S)** | **High susceptibility/ High risk** | **Medium susceptibility/ Medium risk** | **Low susceptibility/ Low risk** |
| | **Rank score = 3** | **Rank score = 2** | **Rank score = 1** |
| S1: Contribution to total catch | >0.2% | 0.04–0.2% | <0.04% |
| S2: Encounterability | High overlap with trawl fishing gear (20 to 60 m depth) | Medium overlap with trawl fishing gear (10 to 20 m depth) | Low overlap with trawl fishing gear (0 to 10 m, >70 m depth) |
| S3: Availability: range of distribution | Limited (western-Pacific) | Widespread (Indo-Pacific) | Global |
| S4: Schooling behavior | Schooling or aggregation | Solitary or schooling or aggregation | Solitary |

## 3. Results

### 3.1. Catch Compositions

After taxonomic classification, all but the beam trawler at sizes S and L had catch compositions with over 100 species (Table 2). The size S otter-board trawler and size L pair trawler showed the most diversity in catch composition—as high as 202 species. Among catch-groups, demersal fish were the most diverse in every trawl type and size; as many as 93 species were recorded for size M otter-board trawlers and size L pair trawlers. The

number of pelagic fishes ranged between seven and 60 species, for size L beam trawlers and size L pair trawlers, respectively. The highest number of elasmobranch species (nine) was found for size M and S otter-board trawlers, while fewer were caught by beam trawlers (three to six species). Of the other (non-fish) aquatic animals, shrimp represented the most diverse group in the catches of trawls, followed by cephalopods and crabs. Meanwhile, other invertebrates not included in our taxonomic classification, such as sea cucumbers, sponges and worms, were a very minimal part of the catch composition. Among all the harvested species, only stingray *Himantura gerrardi* is ranked "endangered" in IUCN's Red List, while the other elasmobranchs are either ranked as "vulnerable" or "near threatened". Among the bony fishes, *Pampus argenteus* is listed as "vulnerable", and *Scomberomorus commerson* and *Diagramma pictum* are "near threatened." None of the other aquatic animals are included in threatened categories. However, it is worth noting that a number of the fishes and other aquatic animals are "data deficient" (https://www.iucnredlist.org (accessed on 18 May 2022); Table S1).

**Table 2.** Number of species of fishes and other aquatic animals caught by trawl fisheries in the Gulf of Thailand and used in this study.

| Group | Otter-Board Trawler | | | | Pair Trawler | | | Beam Trawler | | |
|---|---|---|---|---|---|---|---|---|---|---|
| | SS | S | M | L | M | L | XL | S | M | L |
| Elasmobranch | 4 | 9 | 9 | 8 | 6 | 8 | 6 | 4 | 6 | 3 |
| Demersal fish | 65 | 86 | 93 | 78 | 69 | 93 | 47 | 32 | 48 | 22 |
| Pelagic fish | 40 | 53 | 50 | 40 | 50 | 60 | 38 | 11 | 21 | 7 |
| Cephalopod | 17 | 22 | 21 | 17 | 18 | 19 | 13 | 9 | 11 | 6 |
| Shrimp | 19 | 22 | 18 | 12 | 4 | 13 | 8 | 11 | 15 | 11 |
| Crab | 6 | 7 | 6 | 6 | 6 | 6 | 6 | 5 | 7 | 6 |
| Other invertebrates | 3 | 3 | 3 | 3 | 3 | 3 | 1 | 1 | 1 | 1 |
| **Total** | 154 | 202 | 200 | 164 | 156 | 202 | 119 | 73 | 109 | 56 |

A list of the top ten taxa contributing to the catch, by percentage of total weight for each trawler category, is presented in Figure 1. For the otter-board and pair trawlers, the species were more evenly mixed and dominated by *Leiognathus* spp. and *Encrasicholina* spp., respectively, which accounted for around 15% of the catch. The beam trawlers, in contrast, were more selective to the shrimps *Penaeus merguiensis* and *Metapenaeus affinis*, both of which contributed over 50%. Contribution to catch, as percentage of weight, by group was distinctly different among the three trawl types (Figure 2). Demersal fishes dominated the catches of otter-board trawlers and ranged between ~40% and ~60%; furthermore, the contribution increased with vessel size. The demersal species in the catches from otter board trawlers included *Leiognathus* spp., *Saurida elongata*, *Priacanthus tayenus*, *Gerres filamentosus* and *Upeneus* spp. The contribution of crabs, e.g., *Portunus pelagicus*, and shrimps, e.g., *Metapenaeopsis* spp. and *Metapenaeus affinis*, to otter-board trawler landings was greater than for pair trawlers, in particular the size SS vessel. Pelagic fishes ranked second for otter-board trawlers and contributed about 50% to the catches of pair trawlers, in which the main taxa included *Encrasicholina* spp., *Stolephorus* spp., *Selaroides leptolepis*, *Rastrelliger* spp. and *Sardinella gibbosa*. For the beam trawlers, over 60% of the landed weight was from shrimps, whereas shrimps contributed about 20% of the weight of catches from small otter-board trawlers (size S and SS). Crabs also contributed substantially to the catches of otter-board and beam trawlers (~2% and ~10%, respectively). Catches of crustaceans by the pair trawlers, however, were minimal. Cephalopods represented around 10% of the catch weight from otter-board trawlers, and the figure was slightly higher for pair trawlers. Finally, the weight of elasmobranchs was generally less than 1% of the catch from all trawl types and sizes.

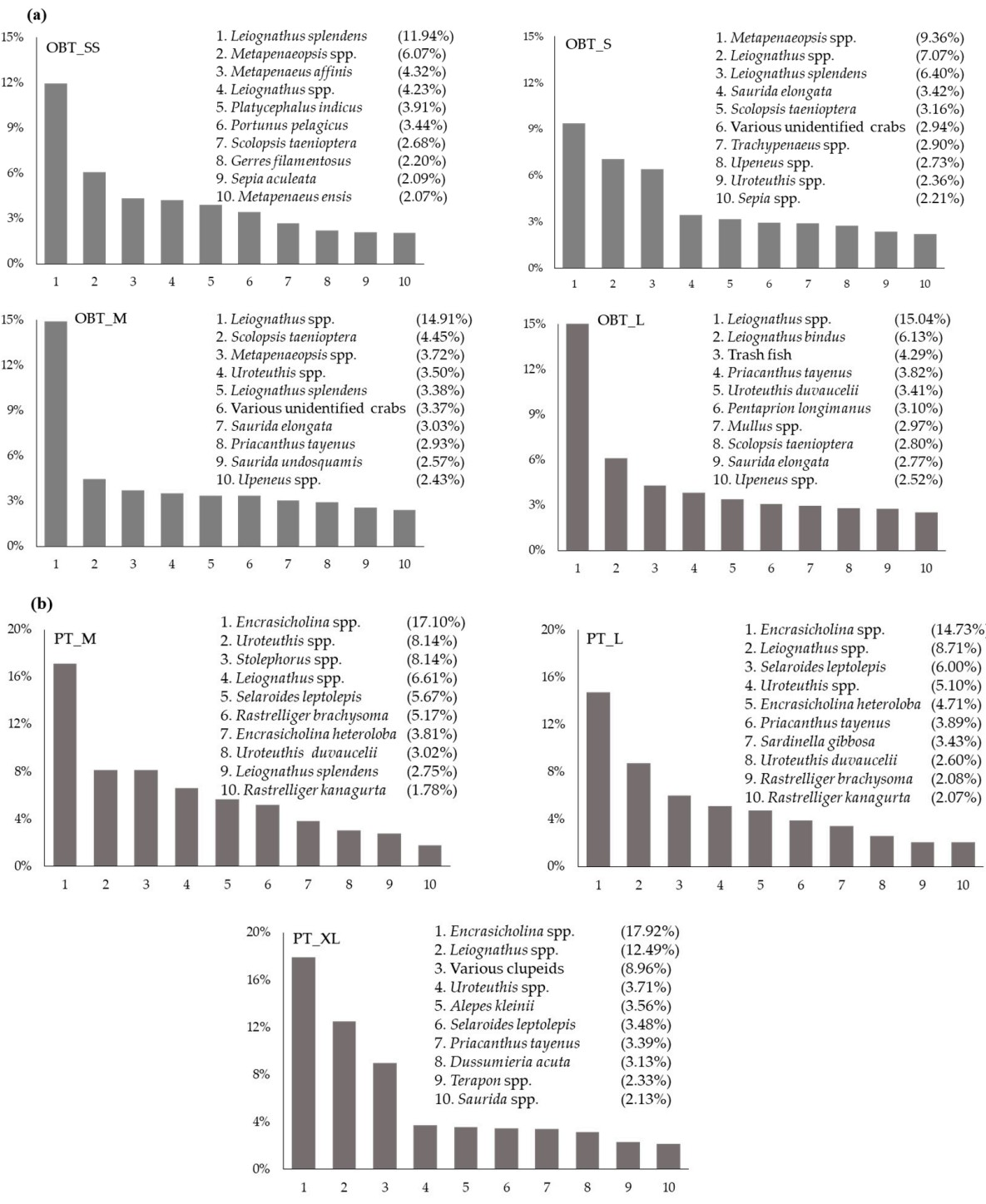

**Figure 1.** *Cont.*

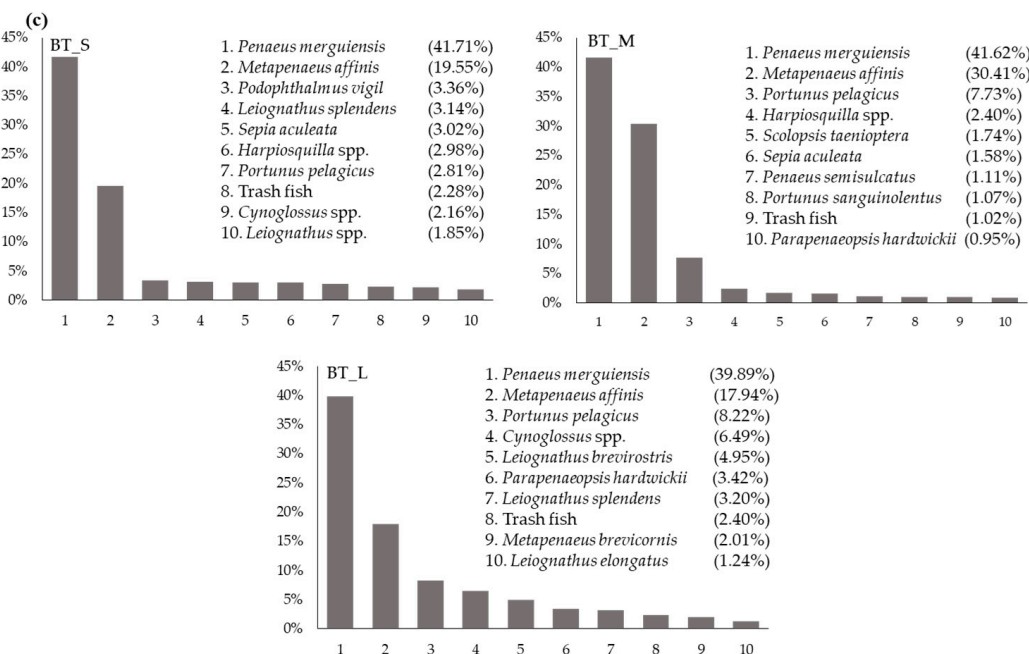

**Figure 1.** Top ten taxa, by weight, in landings of trawl fisheries in the Gulf of Thailand, by trawl type and vessel size. Trawl type: (**a**) OBT = Otter-board, (**b**) PT = Pair trawl and (**c**) BT = Beam trawl. Vessel size: SS (<10 gross tonnage, GT), S (10–29.9 GT), M (30–59.9 GT), L (60–149.9 GT) and XL (>150 GT).

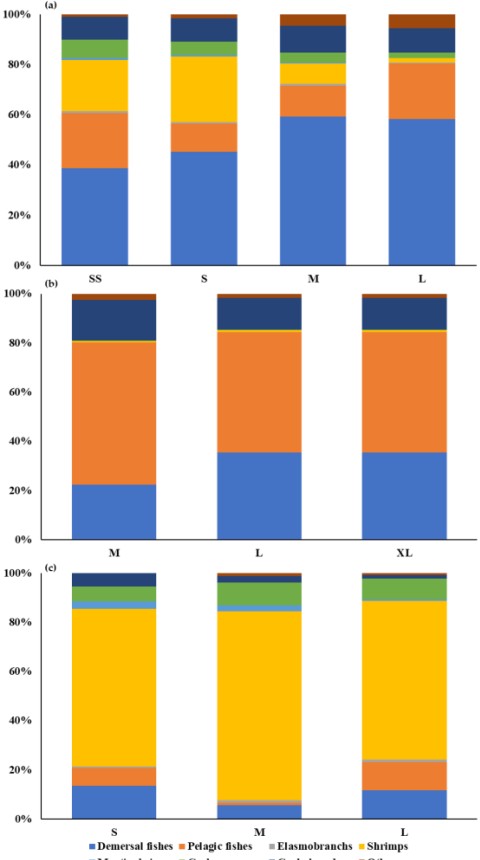

**Figure 2.** Contribution by taxa groups to catches of trawl fisheries in the Gulf of Thailand, by trawl type and vessel size. Trawl type: (**a**) OBT = Otter-board, (**b**) PT = Pair trawl and (**c**) BT = Beam trawl. Vessel size: SS (<10 gross tonnage, GT), S (10–29.9 GT), M (30–59.9 GT), L (60–149.9 GT) and XL (>150 GT).

### 3.2. Vulnerability Analysis

The overall V-score ranged between 1.00 and 3.53 with the average of 2.55 ± 0.30, and none of the mean data quality scores, for either productivity or susceptibility attributes, were beyond 3 (Table S1). The high-vulnerability taxa (V-score ≥ 3.18) were found for all trawl types and sizes, except the beam trawler size S. There were only seven taxa classified as high vulnerability: four taxa of teleosts, namely *Saurida elongata*, *Plotosus* spp., *Gymnothorax* spp. and *Sphyraena* spp. and three taxa of elasmobranchs, namely *Carcharhinus* spp. *Brevitrygon heterura* and *Neotrygon kuhlii*. However, when the high-medium vulnerability taxa (3.00 ≤ V-score < 3.18) were tallied, the total was as high as 26 taxa, for otter-board trawler size L (Table 3). The high-medium vulnerability taxa were not only fishes but also cephalopods; the number of teleost, elasmobranch and cephalopod taxa in this category were 38, five and three species, respectively.

**Table 3.** List of high and high-medium vulnerability species in the trawl fishery in the Gulf of Thailand by each trawl-type and vessel size with its V-score in parentheses. Species with * indicates that they were among the top ten species caught, in terms of weight, by the trawlers.

| (a) Otter-Board Trawler | | |
|---|---|---|
| **Vessel Size** | **High Vulnerability Species** | **High-Medium Vulnerability Species** |
| SS | *Brevitrygon heterura* (3.29), *Saurida elongata* (3.26) and *Plotosus* spp. (3.18) | *Megalops cyprinoides* (3.16), *Apogon* spp (3.13). *Trichiurus lepturus* (3.13), *Platycephalus indicus* * (3.07), *Saurida isarankurai* (3.06), *Gymnothorax* spp. (3.06), *Mugil* spp. (3.02), *Upeneus* spp. (3.00), *Gazza minuta* (3.00), *Nemipterus hexodon* (3.00), *Sillago sihama* (3.00), *Uroteuthis duvaucelii* (3.13), *Uroteuthis chinensis* (3.13) and *Uroteuthis* spp. (3.13) |
| S | *S. elongata* * (3.26), *Plotosus* spp. (3.18), and *Gymnothorax* spp. (3.18) | *Maculabatis gerrardi* (3.17), *B. heterura* (3.08), *Chiloscyllium punctatum* (3.06), *Apogon* spp. (3.13), *P. indicus* (3.07), *Sphyraena* spp. (3.04), *Mugil* spp. (3.02), *Lutjanus lutjanus* (3.00), *N. hexodon* (3.00), *S. isarankurai* (3.00), *Scolopsis taenioptera* * (3.00), *Atule mate* (3.00), *Selaroides leptolepis* (3.00), *U. chinensis* (3.13), *U. duvaucelii* (3.13) and *Uroteuthis* spp. * (3.13) |
| M | *S. elongata* * (3.26), *Gymnothorax* spp. (3.18) and *Sphyraena* spp. (3.18) | *M. gerrardi* (3.17), *B. heterura* (3.08), *Carcharhinus melanopterus* (3.01), *C. punctatum* (3.06), *Apogon* spp. (3.13), *P. indicus* (3.07), *Terapon jarbua* (3.07), *Plotosus* spp. (3.04), *Mugil* spp.(3.00), *L. lutjanus* (3.00), *N. hexodon* (3.00), *Nemipterus marginatus* (3.00), *Nemipterus nemurus* (3.00), *Nemipterus tambuloides* (3.00), *Pentaprion longimanus* (3.00), *S. isarankurai* (3.00), *S. taenioptera* * (3.00), *A. mate* (3.00), *S. leptolepis* (3.00), *U. chinensis* (3.13), *U. duvaucelii* (3.13) and *Uroteuthis* spp. * (3.13) |
| L | *Carcharhinus* spp. (3.18), *S. elongata* * (3.26) and *Sphyraena* spp. (3.18) | *M. gerrardi* (3.17), *C. punctatum* (3.06), *Lutjanus johnii* (3.16), *Lutjanus sebae* (3.16), *Apogon* spp. (3.13), *Megalaspis cordyla* (3.13), *T.l epturus* (3.13), *Cynoglossus* spp. (3.12), *P. indicus* (3.07), *T. jarbua* (3.07), *Chirocentrus dorab* (3.07), *Encrasicholina heteroloba* (3.06), *Gymnothorax* spp. (3.06), *Mugil* spp. (3.02), *L. lutjanus* (3.00), *N. hexodon* (3.00), *N. marginatus* (3.00), *N. tambuloides* (3.00), *P. longimanus* * (3.00), *S. isarankurai* (3.00), *S. taenioptera* * (3.00), *A. mate* (3.00), *S. leptolepis* (3.00), *U. chinensis* * (3.13), *U. duvaucelii* * (3.13) and *Uroteuthis* spp. (3.13) |

**Table 3.** *Cont.*

| (b) Pair Trawler | | |
|---|---|---|
| **Vessel Size** | **High Vulnerability Species** | **High-Medium Vulnerability Species** |
| M | *Plotosus* spp. (3.18) and *S. elongata* (3.26) | *M. gerrardi* (3.10), *Apogon* spp. (3.13), *Scomberomorus commerson* (3.08), *Lutjanus* spp. (3.07), *P. indicus* (3.00), *T. jarbua* (3.07), *C. dorab* (3.07), *Parastromateus niger* (3.06), *E. heteroloba* * (3.06), *Mugil* spp. (3.00), *P. longimanus* (3.00), *L. lutjanus* (3.00), *N. hexodon* (3.00), *N. tambuloides* (3.00), *S. taenioptera* (3.00), *S. leptolepis* * (3.00), *A. mate* (3.00), *Stolephorus indicus* * (3.00), *U. chinensis* (3.13), *U. duvaucelii* * (3.13) and *Uroteuthis* spp. * (3.13) |
| L | *Carcharhinus* spp. (3.38) and *S. elongata* (3.26) | *M. gerrardi* (3.10), *Apogon* spp. (3.13)., *M. cordyla* (3.13), *S. commerson* (3.08), *C. dorab* (3.07), *E. heteroloba* * (3.06), *P. niger* (3.06), *Gymnothorax* spp. (3.06), *Plotosus* spp. (3.04), *Mugil* spp. (3.00), *L. lutjanus* (3.00), *N. hexodon* (3.00), *N. tambuloides* (3.00), *P. longimanus* (3.00), *S. isarankurai* (3.00), *S. taenioptera* (3.00), *A. mate* (3.00), *S. leptolepis* * (3.00), *S. indicus* (3.00), *U. chinensis* (3.13), *U. duvaucelii* * (3.13) and *Uroteuthis* spp.(3.13) |
| XL | *Carcharhinus* spp. (3.53) and *S. elongata* (3.26) | *B. heterura* (3.10), *M. cyprinoides* (3.16), *Apogon* spp. (3.13), *M. cordyla* (3.13), *Pampus argenteus* (3.13), *Diagramma pictum* (3.07), *C. dorab* (3.07), *P. niger* (3.06), *Plotosus* spp. (3.04), *L. lutjanus* (3.00), *P. longimanus* (3.00), *Priacanthus macracanthus* (3.00), *A. mate* (3.00), *S. leptolepis* * (3.00), *U. chinensis* (3.13), *U. duvaucelii* (3.13) and *Uroteuthis* spp. * (3.13) |
| (c) Beam Trawler | | |
| **Vessel Size** | **High Vulnerability Species** | **High-Medium Vulnerability Species** |
| S | None | *M. gerrardi* (3.17), *B. heterura* (3.08), *Muraenesox* spp. (3.08), *Platycephalus* spp. (3.07), *Pomadasys maculatus* (3.07), *T. jarbua* (3.07), *S. elongata* (3.00), *Plotosus* spp. (3.01), *N. hexodon* (3.00), *Saurida* spp. (3.00) and *S. taenioptera* (3.00) |
| M | *B. heterura* (3.23) | *Brevitrygon imbricata* (3.17), *C. punctatum* (3.06) and *Plotosus* spp. (3.04) |
| L | *Dasyatis kuhlii* (3.25) | *B. heterura* (3.03) and *T. jarbua* (3.07) |

Focusing on the ten taxa with highest contribution to the catch, by weight (Table 3), the high-vulnerability *S. elongata* is listed for otter-board trawlers of size S, M and L. Six (6) out of the top ten taxa are ranked as high-medium or highly vulnerable. The demersal fish *Scolopsis taenioptera*, which is ranked as a high-medium vulnerability species, is included for all sizes of otter-board trawlers except SS, as well as for size S beam trawlers. Two (2) pelagic fishes in the top ten taxa caught by pair trawlers, namely *Encrasicholina heteroloba* and *S. leptolepis*, were ranked as high-medium vulnerability. In addition, the loliginid squids *Uroteuthis* spp., which are squids with high-medium vulnerability, were among the top ten taxa in catches by otter-board (size S, M and L) and pair (size M and L) trawlers.

Graphical PSA results for the high and high-medium vulnerability taxa, by each trawl type, are displayed in Figure 3. It is clear that most of these taxa are highly vulnerable due to their susceptibility to the fisheries, i.e., high rank scores along the vertical axis. The elasmobranchs, on the other hand, are highly vulnerable due to their low productivity, indicated by high rank scores along the horizontal axis. Elasmobranchs also showed higher vulnerability to the trawl fisheries than other groups for every trawl type and size ($p < 0.05$, Figure 4); of these, *Carcharhinus* spp. had the highest median V-score (3.43), followed by *Himantura heterura* (3.14) and *Carcharhinus melanopterus* (3.01).

Based on the PSA analysis, catches from trawl fisheries in the GoT were categorized into four (4) distinct categories as high, high-medium, moderate and low vulnerability



species (Table S1). It can be seen that all elasmobranchs were ranked either as high or high-medium vulnerability species. Meanwhile, vulnerability of the teleosts and other aquatic animals mainly depended on their proportion in the catches.

### 3.3. Trend and Variation Analyses in Fish Landings

Since the annual statistical report presents fish landings as both individual species and groups of species, 20 high and high-medium vulnerability taxa were available with long-term data series, i.e., between 1985 and 2020 (Figure 5). No trends were found in two of the taxa, namely *Muraenesox* spp. and cephalopod *Uroteuthis* spp, throughout the considered period. Meanwhile, the remaining 18 taxa showed both continuous and discontinuous monotonic trends in their time series of annual landings. A continuous increase over time was found in *Sphyraena* spp., while a continuous decrease was observed in *S. commerson* and *Mugil* spp. Many types of discontinuous trends were observed. Trend inversion was observed, both as "positive-then-negative" (7 taxa) and "negative-then-positive" sub-series (6 taxa). The former included rays and sharks as well as *Saurida* spp., while the latter included other bony fishes such as *Megalapis cordyla* and *Plotosus* spp. Trend breaks, i.e., with two significant positive sub-series, were seen only in *Lutjanus* spp. The trend for anchovies (i.e., mixed *Stolephorus* spp. and *Encrasicholina* spp.) was very interesting since it nearly showed "positive-then-negative" inversion, but the data from recent years showed an increase followed by a relatively constant level, i.e., no trend.

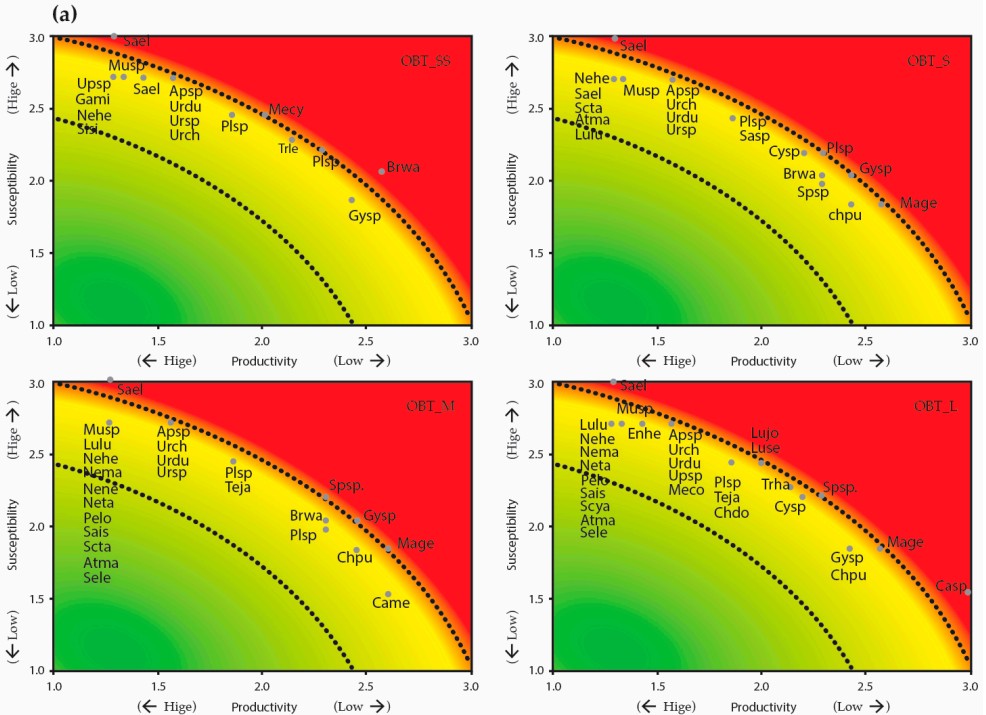

**Figure 3.** *Cont.*

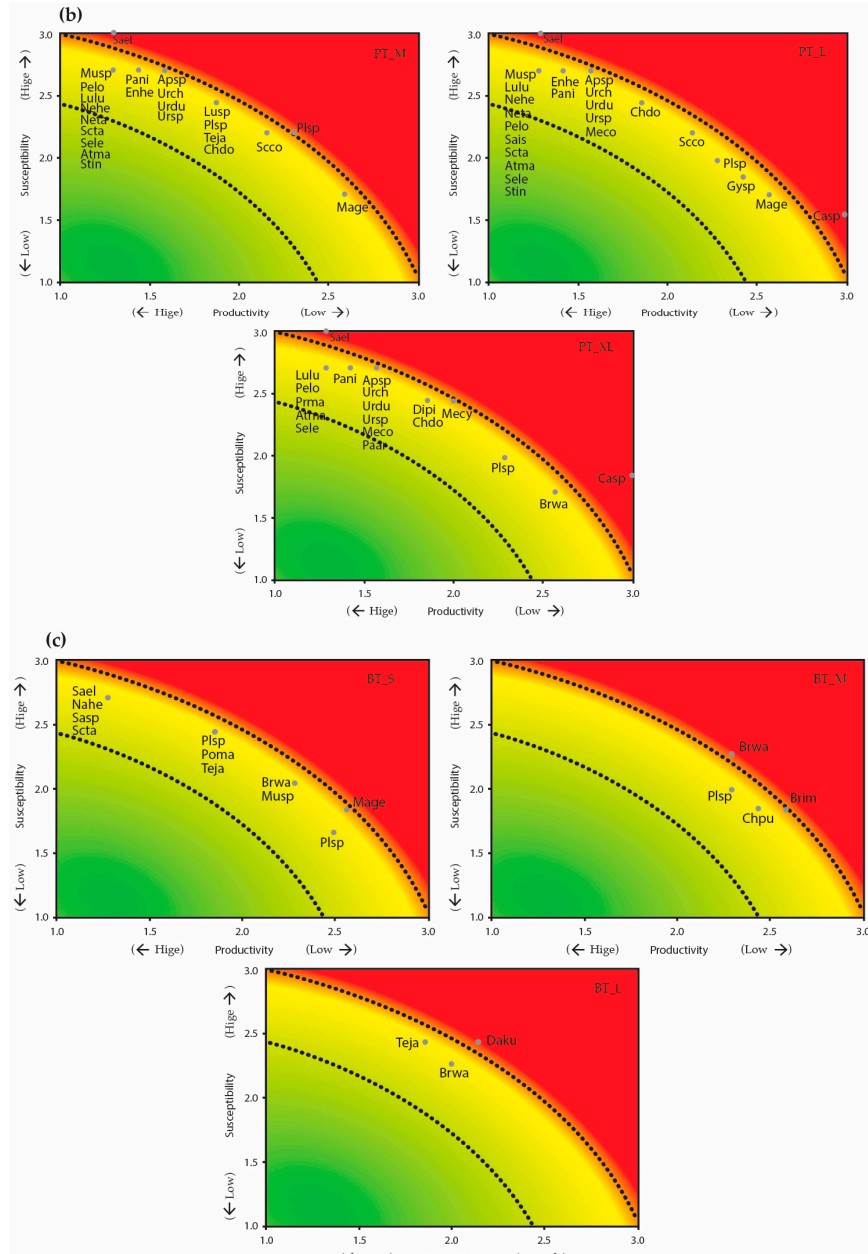

**Figure 3.** Productivity-susceptibility plots of high and high-medium vulnerability taxa in landings of trawl fisheries in the Gulf of Thailand by trawl type and vessel size. Trawl type: (**a**) OBT = Otter-board trawl, (**b**) PT = Pair trawl and (**c**) BT = Beam trawl. Vessel size: SS (<10 gross tonnage, GT), S (10–29.9 GT), M (30–59.9 GT), L (60–149.9 GT) and XL (>150 GT). The curved dotted lines represent boundaries for low vulnerability (V score < 2.64), moderate vulnerability (V score $2.64 \leq V \leq 3.18$) and high vulnerability (V score > 3.18). **Note**: Brwa: *Brevitrygon heterura*, Sael: *Saurida elongata*, Plsp: *Plotosus* spp., Mecy: *Megalops cyprinoides*, Apsp: *Apogon* spp, Gysp: *Gymnothorax* spp., Musp: *Mugil* spp., Trle: *Trichiurus lepturus*, Plin: *Platycephalus indicus*, Sais: *Saurida isarankurai*, Gami: *Gazza minuta*, Nehe: *Nemipterus hexodon*, Sisi: *Sillago sihama*, Urdu: *Uroteuthis duvaucelii*, Urch: *Uroteuthis chinensis*, Ursp: *Uroteuthis* spp., Mage: *Maculabatis gerrardi*, Chpu: *Chiloscyllium punctatum*, Spsp: *Sphyraena* spp., Lulu: *Lutjanus lutjanus*, Scta: *Scolopsis taenioptera*, Came: *Carcharhinus melanopterus*, Teja: *Terapon jarbua*, Nema: *Nemipterus marginatus*, Nene: *Nemipterus nemurus*, Neta: *Nemipterus tambuloides*, Pelo: *Pentaprion longimanus*, Atma: *Atule mate*, Casp: *Carcharhinus* spp., Lujo: *Lutjanus johnii*, Luse: *Lutjanus sebae*, Meco: *Megalaspis cordyla*, Chdo: *Chirocentrus dorab*, Enhe: *Encrasicholina heteroloba*, Scco: *Scomberomorus commerson*, Lusp: *Lutjanus* spp., Pani: *Parastromateus niger*, Prma: *Priacanthus macracanthus*, Musp: *Muraenesox* spp., Plsp: *Platycephalus* spp., Poma: *Pomadasys maculatus*.

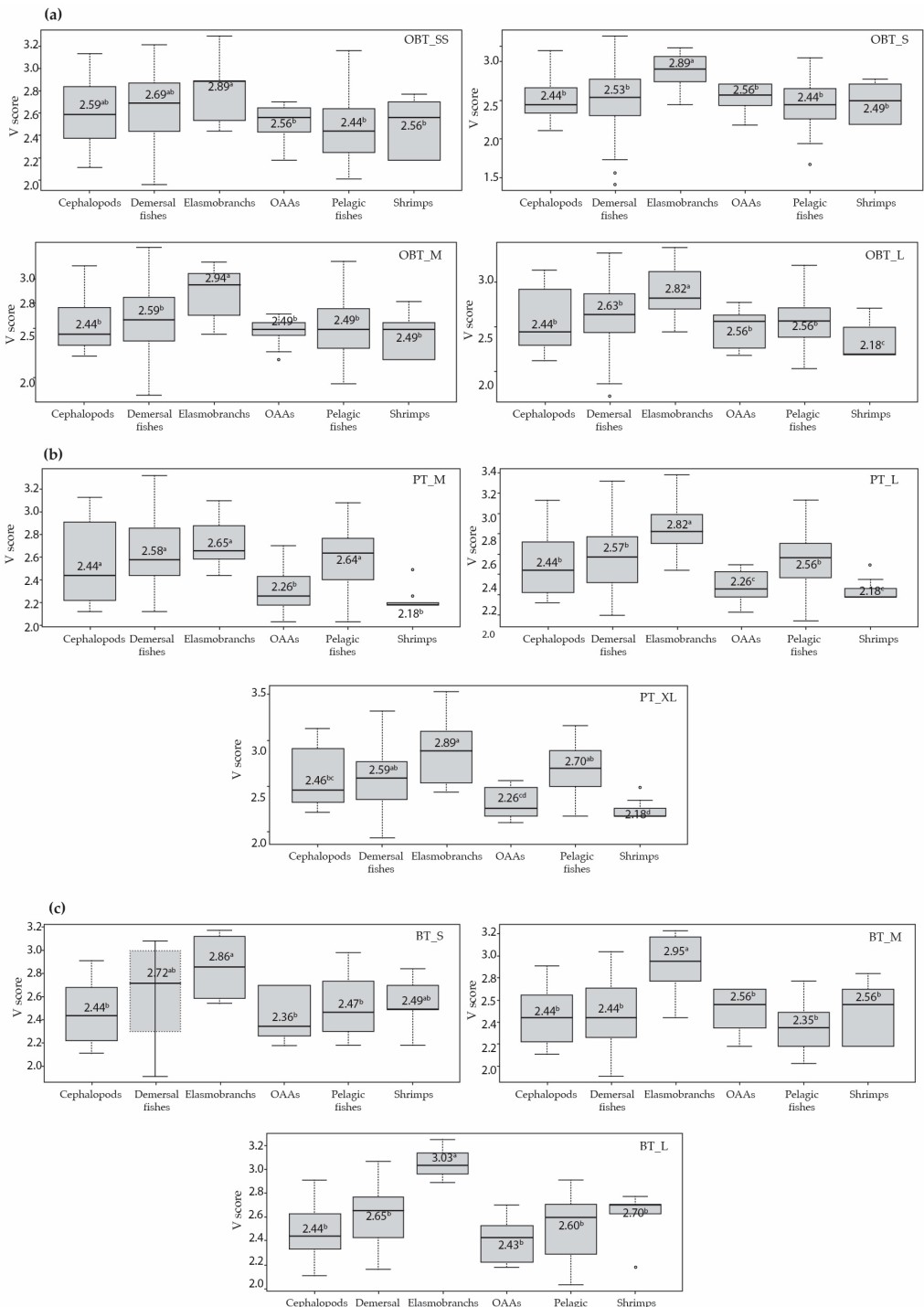

**Figure 4.** Boxplots showing vulnerability scores of taxonomic groups to trawl fisheries in the Gulf of Thailand by trawl type and vessel size. OAAs = Other aquatic animals. Trawl type: (**a**) OBT = Otter-board trawl, (**b**) PT = Pair trawl and (**c**) BT = Beam trawl. Vessel size: SS (<10 gross tonnage, GT), S (10–29.9 GT), M (30–59.9 GT), L (60–149.9 GT) and XL (>150 GT). For each boxplot, the bold line in the center represents the median, the top and bottom of the box represent the first and third quartiles, and the whiskers indicate the highest and lowest values within 1.5× the interquartile range. Median values are presented and compared among groups within the trawl type and vessel size, where different superscripts indicate significant difference (*p* < 0.05) by Dunn's post test.

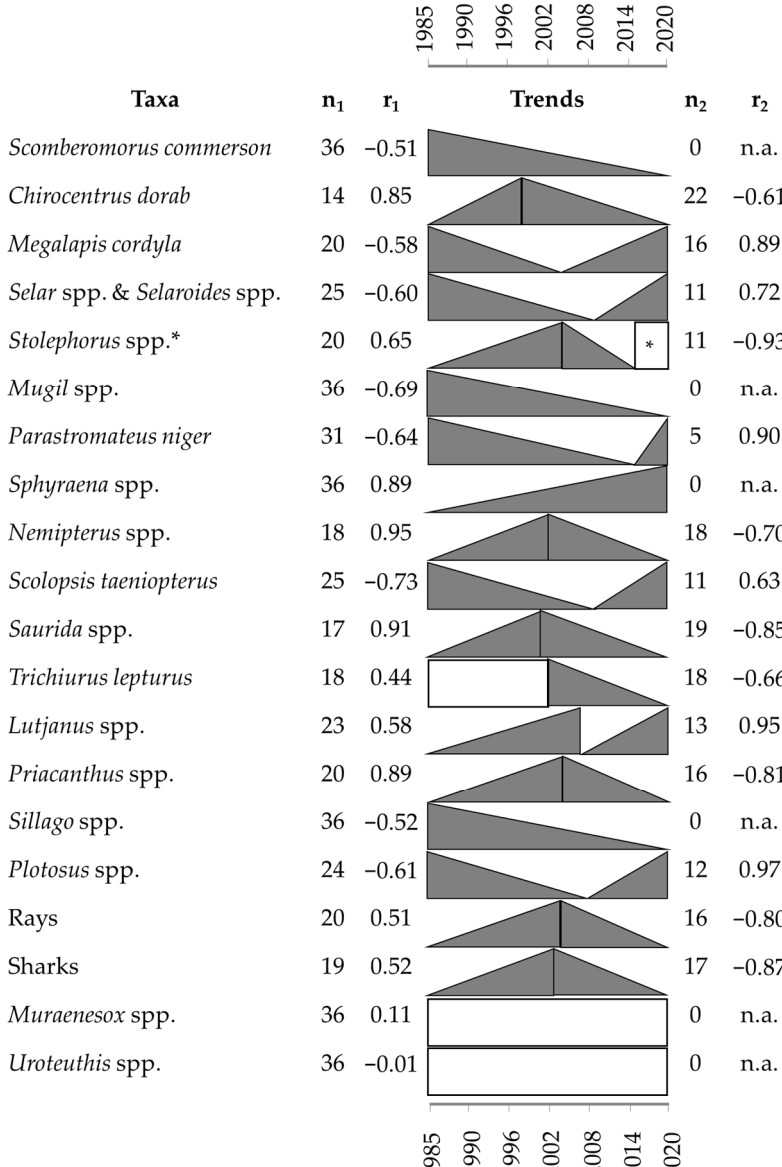

**Figure 5.** Trends in fish landings of the high and high-medium vulnerability taxa from the trawl fisheries in the Gulf of Thailand from 1985 to 2020. From the left: taxa, number of years ($n_1$) and Spearman correlation ($r_1$) for the first subseries, representation of the trend, number of years ($n_2$) and Spearman correlation ($r_2$) for the second subseries. Note * no trend in fish landings has been observed in *Stolephorus* spp. since 2019.

Variations in the fish landings of the high and high-medium vulnerability species or groups of species, based on data available in the fisheries statistics of Thailand, are presented in Table 4. For long-term variation, the modes of coefficients of variation were 30% for CV0 and 10% for CV1 and CV2. The plots between CV0 to CV1 and CV2 show that all the coordinates were below the bisectrix line (Figure 6), confirming there were significant trends in fish landings for our selected taxa. Meanwhile, both the absolute ($U_a$) and relative ($U_r$) inter-annual (i.e., short-term) variation showed positive skew, with modes of 20% and 10%, respectively. The mean value of $U_r$ (24.8% ± 12.3%) was slightly higher than $U_a$ (24.1% ± 11.9%) but not with a significant difference (*t*-test, *p*-value = 0.85).

**Table 4.** Coefficient of variation (%) for zero- (CV0), first- (CV1) and second- (CV2) order polynomials and indices (%) for absolute ($U_a$) and relative ($U_r$) short-time variations in fish landings of high and considerably high vulnerability species/groups of species from trawl fisheries in the Gulf of Thailand.

| Species/Group of Species | CV0 | CV1 | CV2 | $U_a$ | $U_r$ |
|---|---|---|---|---|---|
| *Scomberomorus commerson* | 37.02 | 10.75 | 14.15 | 17.26 | 16.43 |
| *Chirocentrus dorab* | 42.24 | 11.53 | 21.03 | 22.85 | 21.53 |
| *Megalapis cordyla* | 70.20 | 6.69 | 40.46 | 35.02 | 37.24 |
| *Selar* spp. (and *Selaroides* spp.) | 23.13 | 0.23 | 10.88 | 13.48 | 12.88 |
| *Stolephorus* spp. | 17.80 | 2.70 | 11.29 | 10.01 | 10.74 |
| *Mugil* spp. | 25.92 | 18.48 | 19.03 | 15.81 | 15.66 |
| *Parastromateus niger* | 45.50 | 2.82 | 20.73 | 21.68 | 20.80 |
| *Sphyraena* spp. | 57.26 | 50.25 | 50.56 | 20.40 | 19.08 |
| *Nemipterus* spp. | 47.95 | 6.28 | 38.41 | 15.68 | 17.35 |
| *Scolopsis taeniopterus* | 159.89 | 115.31 | 141.99 | 36.24 | 39.00 |
| *Saurida* spp. | 61.07 | 9.82 | 47.66 | 25.02 | 24.40 |
| *Trichiurus lepturus* | 50.99 | 11.20 | 26.00 | 27.87 | 60.50 |
| *Lutjanus* spp. | 52.52 | 4.02 | 17.92 | 29.99 | 31.16 |
| *Priacanthus* spp. | 65.55 | 14.35 | 53.90 | 16.32 | 16.88 |
| *Sillago* spp. | 75.95 | 14.53 | 31.19 | 29.14 | 30.97 |
| *Plotosus* spp. | 133.32 | 37.08 | 40.24 | 63.71 | 40.36 |
| Rays | 57.13 | 27.38 | 35.96 | 24.91 | 22.60 |
| Sharks | 78.05 | 25.67 | 48.61 | 25.65 | 28.21 |
| *Muraenesox* spp. | 32.25 | 1.47 | 31.19 | 20.81 | 20.05 |
| *Uroteuthis* spp. | 10.65 | 1.47 | 1.53 | 10.34 | 9.88 |

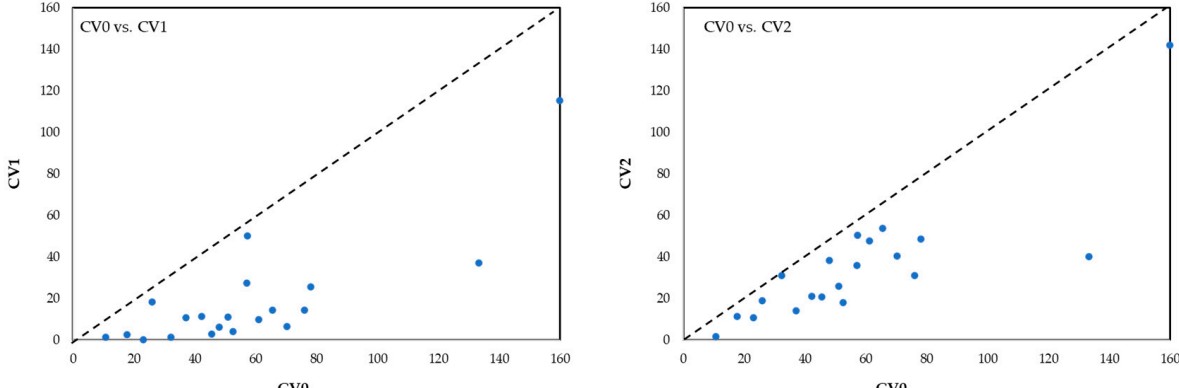

**Figure 6.** Plots between the CV0 and higher order CVs of landings of the high and high-medium vulnerability taxa from the trawl fisheries in the Gulf of Thailand.

## 4. Discussion

Management of the trawl fisheries in Southeast Asia is quite challenging, considering the volume of landings and fishing pressure [10,12]. Approximately 300 species of fishes and other aquatic animals contribute to the catches of trawl fisheries, representing almost all major groups of aquatic animals which have different degrees of resilience and potential for stock recovery [4,9,12,15]. Increasing effort by mixed-species trawl fisheries normally causes a number of target and bycatch species to be overfished and can impair recruitment for the more vulnerable taxa, potentially leading to stock collapse and imbalance in the ecosystem [33,34]. Understanding the catch composition, identifying high or considerably high vulnerability species and determining their stock status and trends are among the most important needs for precautionary mitigation. Only then can suitable management practices be applied to develop more sustainable trawl fisheries in Southeast Asia.

In the Gulf of Thailand (GoT), otter-board trawlers contributed the highest landings among the three trawl types, roughly half the total from all trawlers. Additionally, the catch of trash fishes from size S vessels of this trawler was reportedly high, at times as

high as 60% [35,36]. Our results clearly reveal that the otter-board trawlers targeted mainly demersal species and that the percentage of this group increased with the size of the vessel. Targeting of demersal species by otter-board trawlers was also reported in the Andaman Sea, where *Saurida* spp. was the most common taxon in the catch [37]. Widodo and Mahiswara [38] reported that demersal fishes such as *Leiognathus* spp., *Nemipterus* spp., *Upeneus* spp. and *Arius* sp. were among the main demersal species caught by otter-board trawlers in North Sumatra, Indonesia. The pair trawlers mostly operate 30–60 nautical miles offshore in the GoT, and pelagic fishes such as *Rastrelliger* spp., *Scomberomorus* spp. and *Decapterus* spp. were dominant in their landings; about half of the catch was composed of trash fishes [9,35]. The catches by pair trawlers in Kien Giang, Viet Nam, also contained a high proportion of anchovies and other pelagic fishes [39]. Lastly, catches from the beam trawlers, which contributed less than 5% of the total from all trawl types, were dominated by shrimps, similar to the small otter trawlers in Kien Giang [10,40]. Comparing our results to previous studies suggests that there has been no change in the dominant taxa in the landings by each trawl type since the Reforming of Fisheries Act in 2015 or since controls on fishing effort were first applied in 2016.

*4.1. Risks Posed by Trawl Fisheries*

Potential vulnerability of fish stocks to the fisheries can be assessed by the PSA, although it does not provide the current status of the stocks [16,26,27]. A step forward has been made to include more attributes and refine the steps of analysis, whereby the medium vulnerability species have increased V-scores [18]. However, keeping the fundamental attributes and analysis as suggested by FAO [12] allows us, for the first time, to screen the stocks with high risk from trawl fisheries in the GoT. The rank scores of productivity attributes of most of teleost fishes caught by the trawl fisheries in the GoT were either 1 or 2, indicating their high productivity; similar results were obtained from other tropical fish stocks [20,26,27]. However, high rank scores of productivity attributes could be found in some exceptions in the species with long life span and high trophic level such as *Scomberomorus commerson*, *Sphyraena* spp., *Plotosus* spp. and *Gymnothorax* spp. In general, two primary groups of fishes can be distinguished according to productivity attributes viz., larger, longer-lived species and smaller, shorter-lived species, where the latter tend to be at lower risk from fisheries [16,17]. However, high fluctuation in interannual recruitment could be observed for some shorter-lived species, though they have higher productivity scores [28]. It can be concluded, therefore, that the risk presented by the trawl fisheries to most of the teleost fishes was due to their susceptibility to capture. Although the attributes related to gear operation per se, e.g., selectivity and post-capture mortality [12], were not applied to PSA in this study due to lack of data, stakeholders all agreed during our discussions that with the cod-end mesh size of 4 cm stretched mesh, all trawlers in the GoT were indiscriminate to both type and size of animals captured. Meanwhile, based on the experience of the participants, the post-capture mortality was relatively high for any escaped teleost fishes, though with variation according to size and species. Suuronen [41] mentioned that any fishes escaping from the cod-end would have increased post-escape mortality and that increasing cod-end mesh size would automatically reduce the injury and mortality of escaping fish. Among the highly vulnerable stocks of teleost fishes, quantitative stock assessment had previously been conducted only for *S. elongata*. The results showed the exploitation rate to be around 0.70, which is beyond the optimum level of 0.5 [42]. Risk to economically important species in the trash fish category should be also considered, as their proportion in the landings of trawl fisheries in the GoT was estimated to be about 15–17% of the total [43].

The elasmobranchs, though contributing less than 1% of the catch by weight, should receive special attention since they are classified as threatened and also show higher vulnerability (V-score) than other groups in the trawl fisheries in the GoT. High vulnerability of the elasmobranchs is largely due to their low productivity relative to the teleost fishes and is strongly affected by age at maturity [44,45]. Coastal trawl fisheries generally showed

higher elasmobranch bycatch ratios than offshore fisheries [46]. Similarly, our results showed higher percentage of elasmobranchs in catches from beam trawlers, since these vessels targeted shrimps and were more likely to operate near the shore [45,47]. A recent study on the stock status of the commonly caught *Chiloscyllium* spp. elsewhere in Southeast Asia revealed alarming rates of overfishing in many stocks, including the GoT, though here it was not yet at the level of growth overfishing [48,49]. To address this, the DOF has launched the Thailand National Plan of Action for the Conservation and Management of Sharks (Plan 1, 2020–2024), ensuring both conservation and sustainable management of sharks in Thai waters [50]. The findings of this study can, therefore, prioritize the species that require an in-depth study in stock status for implementing effective science-based management plans.

The squids *Uroteuthis* spp. were also shown to have high vulnerability from the trawl fisheries in the GoT since they contributed substantially to the landings [51]. Utilization of this group shows increasing trends globally, but stock assessments are still limited [52,53]. Kongprom et al. [54] reported that the exploitation rates of *U. chinensis* and *U. duvaucelii* in the GoT were 0.52 and 0.56, respectively, and about 10% over the optimum fishing effort at maximum sustainable yield for these stocks. Among the other aquatic animals targeted by the beam trawlers, the penaeid shrimps still show low vulnerability in this fishery because of their high resilience, due to factors such as high fecundity and short life span [55]. Low vulnerability of other aquatic animals to the GoT trawl fisheries were similarly observed elsewhere in Thailand [19,20]. However, it must be emphasized that this PSA is only a first step in evaluating stock status, and a more definitive analysis should be conducted as more data become available [16,56]. For example, Sin-anun and Kwanchai [57] reported that the stock of shrimp *Trachypenaeus fulvus* in the GoT was over-exploited, though the shrimp group as a whole was shown to have low vulnerability in this analysis.

*4.2. Trend and Variation in Fish Landings*

Although the accuracy of reported marine fish landings has often been questioned and data reconstruction is required for many of the world's fisheries, including the GoT [4,58,59], the official harvest data can at least provide a picture of the trends and variation in stocks of target species or groups of species, which relate directly to their abundance in the fishing grounds [30,31]. Increasing or decreasing trends in fish landings could be also affected by fishing effort and market demand [29]. For example, estimates in China revealed that significant increases in domestic marine catches were highly correlated with escalating fishing power [60]. Freire et al. [59] mentioned that changes in the catch of any individual species in a fishing ground should show similar trends for both artisanal and commercial fisheries.

Caution could be raised for the species or groups of species with a continuously decreasing trend or a "positive-then-negative" trend inversion. These trends were seen in predatory teleosts, e.g., *S. commerson*, *Mugil* spp. and *Suarida* spp., which normally live longer and have lower resilience than other species. Suuronen et al. [10] suggested that the trawl fisheries in SEA countries switch to targeting a number of small, shorter-lived and fast-recruiting species to maintain their catch and profit levels. A declining trend in landings of sharks and rays in the GoT is in line with the global trend for these fishes [47], though comprehensive catch data for elasmobranchs are still limited [53]. Dureuil et al. [61] also showed a tremendous decline in abundance of elasmobranchs in heavily trawled areas in European waters. Meanwhile, increasing trends in stocks of many species or groups of species in recent years, as well the stabilization seen in *Stolepholus* spp., could be attributed to the strict controls on fishing effort by the DoF. It has already demonstrated elsewhere that appropriate management measures, under science-based fisheries recommendations, can improve the stock status [62]. The continuously increasing trend of *Sphyraena* spp. landings, however, should be closely monitored, with focus on size distribution of the catch and catch per unit effort, as low values of both risk indices imply possible overexploitation [63].

The lack of trends in fish landings of *Muraenesox* spp. and *Uroteuthis* spp. were confirmed by low long-term variation. There is no clear explanation for *Muraenesox* spp., but Chotiyaputta et al. [51] mentioned that catches of cephalopods remain relatively constant in the GoT due to their life history traits of rapid turnover and lower standing stocks. The short-term variations of around 10–20% in most of species and groups imply that the fishers experienced gradual changes and slight year-by-year fluctuations in their catches [31]. For short-term variation, most of the studied taxa showed higher relative annual variation ($U_r$) than absolute variation ($U_a$); thus, it can be interpreted that the fish landings were inversely related to yield, resulting in higher uncertainty when catches are low [30,64].

## 5. Conclusions

This study, for the first time, performs an assessment of the risk posed by trawl fisheries in the GoT using PSA. The assessment covers as many species and groups of species in the catches as possible, which are targeted differently based on trawler type and size. It reveals that vulnerability of different taxa can be distinguishable by the selected attributes for both productivity and susceptibility, as suggested by FAO [12] for PSA study in tropical trawl fisheries. More than 20 teleost fishes, including sharks and rays as well as squids, were classified as vulnerable taxa. The data quality was rated as "moderate", so further studies are recommended to improve the estimated vulnerability scores applied by PSA. Moreover, a comprehensive quantitative stock assessment of these vulnerable taxa should be performed for more certainty of their stock status. Additionally, from long-term time series data of fish landings available for some highly vulnerable taxa, significant trends were observed in all except *Muraenesox* spp. and squids, due to their low long- and short-term variation. This study has identified high-risk species and species groups, which require a precautionary approach for fisheries management and warrant specific management measures and limit reference points. Without these safeguards, fishing pressure may reach a level where recruitment is impaired, which may consequently to an imbalance of the ecosystem. These results are among the first stepping stones on the path to sustainable mixed-species trawl fisheries in the GoT.

**Supplementary Materials:** The following supporting information can be downloaded at: https://www.mdpi.com/article/10.3390/fishes8040177/s1, Table S1: List of species/group of species, their score for each attribute and data quality score used in Productivity and Susceptibility Analysis for the trawl fisheries in the Gulf of Thailand. The vulnerability score (V score) is presented by trawl type and vessel size.

**Author Contributions:** Conceptualization and methodology P.N., S.K. and T.J.; validation, investigation and formal analysis, P.N., S.K., W.T., P.A., R.P. and T.J.; writing—original draft preparation, P.N. and T.J.; writing—review and editing and funding acquisition, T.J. All authors have read and agreed to the published version of the manuscript.

**Funding:** This research was funded by the Agricultural Research Development Agency (ARDA), grant number POP6405032250 and POP6505032110 and The APC was funded by ARDA.

**Institutional Review Board Statement:** Not applicable.

**Data Availability Statement:** The data that support the findings of this study are available from the corresponding author upon request.

**Acknowledgments:** We are grateful to the Department of Fisheries for allowing us to access the original data and the Thai Sustainable Fisheries Roundtable for organizing the stakeholder meeting.

**Conflicts of Interest:** The authors declare no conflict of interest.

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
