# Peer review of "Trawl Fisheries in the Gulf of Thailand: Vulnerability Assessment and Trend Analysis of the Fish Landings"

_fishes, doi:10.3390/fishes8040177_

Round 1

Reviewer 1 Report

The analysis is scientifically sound. The paper is generally well written. Minor editorial revisions and language corrections are needed in places.

Specific comments:

Lines 56 and 118: Change GOT as GoT throughout the MS.

Lines 133 and 208: Change ‘taxonomical’ as ‘taxonomic’

Lines 210 - 213: Italicize all the scientific names, e.g., Himantura gerrardi, Pampus argenteus, Scomberomorus commerson, Diagramma pictum etc.

Marked MS is attached herewith.

Author Response

Reviewer #1 (All revisions, according to Reviewer #1’s comments are yellow highlights)

No.

Comments

Response and Rebuttal

1

The analysis is scientifically sound. The paper is generally well written. Minor editorial revisions and language corrections are needed in places.

The English of the MS was, again, edited by the academic English editing service.

2

Lines 56 and 118: Change GOT as GoT throughout the MS.

The abbreviation “GoT” has been used throughout the manuscript.

3

Lines 133 and 208: Change ‘taxonomical’ as ‘taxonomic’

Done (Lines 144 and 229)

4

Lines 210 - 213: Italicize all the scientific names, e.g., Himantura gerrardi, Pampus argenteus, Scomberomorus commerson, Diagramma pictum etc

Scientific names were italicized throughout the manuscript

Reviewer 2 Report

page 1, line(s) 44: Please confirm that "(estimated at only 1 %)" is written in the correct position.

Page 2, line(s) 47: Does the author mean the Department of Fisheries, Thailand?

page 3, line(s) 116: I understand that it is challenging to perform the stock assessment on all "individual species" in the trawl fishery, and it is acceptable to include some samples in the "group of species." However, it is suggested that the authors explain the importance of "group of species analysis", which needs to be supported by references.

Page 3, line(s) 133: Because this study included five research centers, the authors should describe how they ensured consistent criteria for identifying species.

Page 4, line(s) 161: From the authors' description, we know that P is a positive indicator and S is a negative one. I need help understanding why V is obtained by adding the square of the two parameters and then opening the root sign. Please explain to the authors.

Page 5, line(s) 188: The authors are requested to confirm the necessity of the mean in the equation.

Page 5, line(s) 193: why "i=2"?

Page 5, line(s) 194: why the "duration of time series data" unit is tonnes.

Page 5, line(s) 195: Please explain to the authors: Which packages are used in R?

Page 5, line(s) 176-195: Please clarify this statistical analysis's principle and reorganize this chapter's text. According to the current version, readers must refrain from repeating this analysis.

Page 5, line(s) 211: I understand that the IUCN Red List is important, but this is different from the research the reader expected to see. And it should be mentioned in the chapter on materials methods.

Page 6, line(s) 219-236: This study holds valuable data; it is a pity that the authors could not present these data brilliantly. Authors are invited to revise these writings to increase the interest of readers.

Page 9, line(s) 280: It is suggested that the authors add a paragraph at the end to summarize the essential information.

Page 11, line(s) 288-290: The authors need to improve the size and resolution of figures 3 and 4.

Page 15, line(s) 351: Figure 5 could be better, and the picture has the characteristics of a table. The authors are advised to improve them. In addition, there is only one trend, which represents the sum of the two groups or a bunch of them? Also, Trends is drawn too casually and lacks scientific drawing standards.

Page 15, line(s) 352-355: Readers still need to spend a lot of time thinking before fully understanding the individual meanings in the tabular data. Suggest the authors improve it.

Page 16, line(s) 359: The authors should strengthen the discussion of the methods of this study and compare it with other studies.

Page 17, line(s) 433-435: It is recommended that the authors describe this study's specific contribution to Thailand's national policy and the SDGs, as mentioned earlier.

Authors are reminded of the consistency of abbreviations, such as DOF and DoF, GOT, and GoT.

Author Response

Reviewer #2 (All revisions, according to Reviewer #2’s comments are green highlights)

No.

Comments

Response and Rebuttal

1

Page 1, line(s) 44: Please confirm that "(estimated at only 1 %)" is written in the correct position.

The only 1% true discard is confirmed (Line 45)

2

Page 2, line(s) 47: Does the author mean the Department of Fisheries, Thailand?

Clarification was made (Line 48)

3

Page 3, line(s) 116: I understand that it is challenging to perform the stock assessment on all "individual species" in the trawl fishery, and it is acceptable to include some samples in the "group of species." However, it is suggested that the authors explain the importance of "group of species analysis", which needs to be supported by references.

Clarification was made (Lines 169-171)

4

Page 3, line(s) 133: Because this study included five research centers, the authors should describe how they ensured consistent criteria for identifying species.

Clarification was made (Lines 144-145)

5

Page 4, line(s) 161: From the authors' description, we know that P is a positive indicator and S is a negative one. I need help understanding why V is obtained by adding the square of the two parameters and then opening the root sign. Please explain to the authors.

Clarification on the estimation of V-score was made (Lines 175-176)

6

Page 5, line(s) 188: The authors are requested to confirm the necessity of the mean in the equation.

The “mean” in the equation is confirmed (Line 209)

7

Page 5, line(s) 193: why "i=2"?

i = 2 is confirmed (Line 214)

8

Page 5, line(s) 194: why the "duration of time series data" unit is tonnes.

Correction (years) was made (Line 215)

9

Page 5, line(s) 195: Please explain to the authors: Which packages are used in R? No any extra package is installed and worked on

We work on the “base” package of R statistics, i.e. no any specific package was used.

10

Page 5, line(s) 176-195: Please clarify this statistical analysis's principle and reorganize this chapter's text. According to the current version, readers must refrain from repeating this analysis.

Text of the sub-section 2.3 “Statistical analysis” was re-written (Lines 193 – 216)

11

Page 5, line(s) 211: I understand that the IUCN Red List is important, but this is different from the research the reader expected to see. And it should be mentioned in the chapter on materials methods.

Mentioning on the IUCN Red List was added in the “Materials and Methods” (Lines 147-148). As well the reference was added, and the series of references were also added accordingly.

12

Page 6, line(s) 219-236: This study holds valuable data; it is a pity that the authors could not present these data brilliantly. Authors are invited to revise these writings to increase the interest of readers.

The content of this section was re-written (Lines 248-255)

13

Page 9, line(s) 280: It is suggested that the authors add a paragraph at the end to summarize the essential information.

A summarized paragraph was already added (Lines 305-309)

14

Page 11, line(s) 288-290: The authors need to improve the size and resolution of figures 3 and 4.

Figures 3 and 4 were already modified

15

Page 15, line(s) 351: Figure 5 could be better, and the picture has the characteristics of a table. The authors are advised to improve them. In addition, there is only one trend, which represents the sum of the two groups or a bunch of them? Also, Trends is drawn too casually and lacks scientific drawing standards.

Figure 5 was already modified

16

Page 15, line(s) 352-355: Readers still need to spend a lot of time thinking before fully understanding the individual meanings in the tabular data. Suggest the authors improve it.

Figure caption of Fig. 5 was revised

17

Page 16, line(s) 359: The authors should strengthen the discussion of the methods of this study and compare it with other studies.

Discussion about the methods was added in each sub-section (Lines 421 -426 and Lines 486-488)

18

Page 17, line(s) 433-435: It is recommended that the authors describe this study's specific contribution to Thailand's national policy and the SDGs, as mentioned earlier.

Specific contribution of the findings to the national policy and SDG was added (Lines 465-467)

19

Authors are reminded of the consistency of abbreviations, such as DOF and DoF, GOT, and GoT.

Done throughout the manuscript

Reviewer 3 Report

General Comment: 

The scientific name of the species should be written in italics, 

Overall, the manuscript needs to be revised for clarity and completeness. The following specific comments and suggestions to improve the manuscript:

Abstract:

It is necessary to add the implications of this study for the management of trawl fisheries.

Introduction:

  1. The introduction should provide more background information about trawl fisheries and their ecological impact. 
  2. Please explain the current progress of the FIP of trawl fisheries in the study area.
  3. Please add information on the general environmental condition in the fishing areas.

Methods:

  1. A map showing the study area and indicating the fishing grounds for different trawl types and vessel sizes should be included to provide spatial context for the study. 
  2. Please provide more detail about the sampling technique and the number of vessels sampled based on trawl types and vessel sizes. 
  3. The taxonomic references used for sample identification should be clearly stated.
  4. The number and composition of discarded biota should be explained, and how this study deals with discards. 

Results:

  1. The data analysis results in the supplement table should be carefully checked (double check) and reported accurately in the manuscript. The biomass data should also be included in the supplement and results section because it explains in the method part.
  2. Please explain the result of the statistical analysis/test stated in the data analysis method, e.g. significantly different test 

Discussion:

1. Please add the implications of the study for trawl fisheries management and ecological impact. It should also discuss the limitations of the study and suggest directions for future research.

2. The bycatch and discards on the results should be discussed in this section.

Conclusion:

The conclusion should summarize the main findings of the study and emphasize the ecological impact of trawl fisheries. It should also provide recommendations for trawl fisheries management based on the study's results.

Author Response

Response and Rebuttals to the Reviewers: fishes-2246674

Reviewer #3 (All revisions, according to Reviewer #3’s comments are blue highlights)

No.

Comments

Response and Rebuttal

1

The scientific name of the species should be written in italics

Done throughout the manuscript.

2

It is necessary to add the implications of this study for the management of trawl fisheries.

Done (Lines 32-33)

3

The introduction should provide more background information about trawl fisheries and their ecological impact.

Background information on ecological impacts were added (Lines 92-94)

4

Please explain the current progress of the FIP of trawl fisheries in the study area.

Current progress of the FIP of trawl fisheries was added (Lines 188-189)

5

Please add information on the general environmental condition in the fishing areas.

General environmental condition in the GoT was added (Lines 57-62)

6

A map showing the study area and indicating the fishing grounds for different trawl types and vessel sizes should be included to provide spatial context for the study

The map of the Thailand’s EEZ within GOT is can google and find elsewhere. Moreover, as the fishing ground of trawl types and sizes are quite mixed and make difficult to be clarify in a single map.

7

Please provide more detail about the sampling technique and the number of vessels sampled based on trawl types and vessel sizes

Done (Lines 139–143)

8

The taxonomic references used for sample identification should be clearly stated

Done (Lines 145-146)

9

The number and composition of discarded biota should be explained, and how this study deals with discards.

As mention in the Introduction that the discard is very minimal, and it is beyond the scope of this study since we focus on the stock, which does not matter that it is main targets or bycatches to understand vulnerability of each stock.

10

The data analysis results in the supplement table should be carefully checked and reported accurately in the manuscript. The biomass data should also be included in the supplement and results section because it explains in the method part.

We did carefully check as suggested by the reviewer. Biomass of each taxa was not collected as we retrieve the data as in percentage of catch, which can be obtained from https://www4.fisheries.go.th/local

/index.php/main/site/strategy-stat

11

Please explain the result of the statistical analysis/test stated in the data analysis method, e.g. significantly different test

We present the P-value in text (Line 304), indeed, we did explain the results in the figure caption of Fig. 4

12

Please add the implications of the study for trawl fisheries management and ecological impact. It should also discuss the limitations of the study and suggest directions for future research.

We did already in the Discussion (Lines 535-538)

13

The bycatch and discards on the results should be discussed in this section

As rebuttal in #9

14

The conclusion should summarize the main findings of the study and emphasize the ecological impact of trawl fisheries. It should also provide recommendations for trawl fisheries management based on the study's results.

We did revise the Conclusion

Round 2

Reviewer 2 Report

The authors had done their best to correct my majority opinion. However, I can't understand the author's response regarding the seventh review comment. In addition, although the authors had worked hard to correct Figures 3 and 4. The authors may need to provide some technical support because there still doesn't seem to be of publishable quality.

Author Response

Response and Rebuttals to the Reviewers: fishes-2246674

Reviewer #2 (All revisions, according to Reviewer #2’s comments are green highlights)

No.

Comments

Response and Rebuttal

1

Page 1, line(s) 44: Please confirm that "(estimated at only 1 %)" is written in the correct position.

The only 1% true discard is confirmed (Line 45)

2

Page 2, line(s) 47: Does the author mean the Department of Fisheries, Thailand?

Clarification was made (Line 48)

3

Page 3, line(s) 116: I understand that it is challenging to perform the stock assessment on all "individual species" in the trawl fishery, and it is acceptable to include some samples in the "group of species." However, it is suggested that the authors explain the importance of "group of species analysis", which needs to be supported by references.

Clarification was made (Lines 169-171)

4

Page 3, line(s) 133: Because this study included five research centers, the authors should describe how they ensured consistent criteria for identifying species.

Clarification was made (Lines 144-145)

5

Page 4, line(s) 161: From the authors' description, we know that P is a positive indicator and S is a negative one. I need help understanding why V is obtained by adding the square of the two parameters and then opening the root sign. Please explain to the authors.

Clarification on the estimation of V-score was made (Lines 175-176)

6

Page 5, line(s) 188: The authors are requested to confirm the necessity of the mean in the equation.

The “mean” in the equation is confirmed (Line 209)

7

Page 5, line(s) 193: why "i=2"?

i = 2 is confirmed (Line 214), the equation is logarithm of the ratio between landing of ith year divide by landing of the previous year. So, if i =1, then denominator is year #0, which cannot estimate.

8

Page 5, line(s) 194: why the "duration of time series data" unit is tonnes.

Correction (years) was made (Line 215)

9

Page 5, line(s) 195: Please explain to the authors: Which packages are used in R? No any extra package is installed and worked on

We work on the “base” package of R statistics, i.e. no any specific package was used.

10

Page 5, line(s) 176-195: Please clarify this statistical analysis's principle and reorganize this chapter's text. According to the current version, readers must refrain from repeating this analysis.

Text of the sub-section 2.3 “Statistical analysis” was re-written (Lines 193 – 216)

11

Page 5, line(s) 211: I understand that the IUCN Red List is important, but this is different from the research the reader expected to see. And it should be mentioned in the chapter on materials methods.

Mentioning on the IUCN Red List was added in the “Materials and Methods” (Lines 147-148). As well the reference was added, and the series of references were also added accordingly.

12

Page 6, line(s) 219-236: This study holds valuable data; it is a pity that the authors could not present these data brilliantly. Authors are invited to revise these writings to increase the interest of readers.

The content of this section was re-written (Lines 248-255)

13

Page 9, line(s) 280: It is suggested that the authors add a paragraph at the end to summarize the essential information.

A summarized paragraph was already added (Lines 305-309)

14

Page 11, line(s) 288-290: The authors need to improve the size and resolution of figures 3 and 4.

Figures 3 and 4 were already modified

15

Page 15, line(s) 351: Figure 5 could be better, and the picture has the characteristics of a table. The authors are advised to improve them. In addition, there is only one trend, which represents the sum of the two groups or a bunch of them? Also, Trends is drawn too casually and lacks scientific drawing standards.

Figure 5 was already modified

16

Page 15, line(s) 352-355: Readers still need to spend a lot of time thinking before fully understanding the individual meanings in the tabular data. Suggest the authors improve it.

Figure caption of Fig. 5 was revised

17

Page 16, line(s) 359: The authors should strengthen the discussion of the methods of this study and compare it with other studies.

Discussion about the methods was added in each sub-section (Lines 421 -426 and Lines 486-488)

18

Page 17, line(s) 433-435: It is recommended that the authors describe this study's specific contribution to Thailand's national policy and the SDGs, as mentioned earlier.

Specific contribution of the findings to the national policy and SDG was added (Lines 465-467)

19

Authors are reminded of the consistency of abbreviations, such as DOF and DoF, GOT, and GoT.

Done throughout the manuscript
